# Influence of Antisynthetase Antibodies Specificities on Antisynthetase Syndrome Clinical Spectrum Time Course

**DOI:** 10.3390/jcm8112013

**Published:** 2019-11-18

**Authors:** Lorenzo Cavagna, Ernesto Trallero-Araguás, Federica Meloni, Ilaria Cavazzana, Jorge Rojas-Serrano, Eugen Feist, Giovanni Zanframundo, Valentina Morandi, Alain Meyer, Jose Antonio Pereira da Silva, Carlo Jorge Matos Costa, Oyvind Molberg, Helena Andersson, Veronica Codullo, Marta Mosca, Simone Barsotti, Rossella Neri, Carlo Scirè, Marcello Govoni, Federica Furini, Francisco Javier Lopez-Longo, Julia Martinez-Barrio, Udo Schneider, Hanns-Martin Lorenz, Andrea Doria, Anna Ghirardello, Norberto Ortego-Centeno, Marco Confalonieri, Paola Tomietto, Nicolò Pipitone, Ana Belen Rodriguez Cambron, María Ángeles Blázquez Cañamero, Reinhard Edmund Voll, Sarah Wendel, Salvatore Scarpato, Francois Maurier, Massimiliano Limonta, Paolo Colombelli, Margherita Giannini, Bernard Geny, Eugenio Arrigoni, Elena Bravi, Paola Migliorini, Alessandro Mathieu, Matteo Piga, Ulrich Drott, Christiane Delbrueck, Jutta Bauhammer, Giovanni Cagnotto, Carlo Vancheri, Gianluca Sambataro, Ellen De Langhe, Pier Paolo Sainaghi, Cristina Monti, Francesca Gigli Berzolari, Mariaeva Romano, Francesco Bonella, Christof Specker, Andreas Schwarting, Ignacio Villa Blanco, Carlo Selmi, Angela Ceribelli, Laura Nuno, Antonio Mera-Varela, Nair Perez Gomez, Enrico Fusaro, Simone Parisi, Luigi Sinigaglia, Nicoletta Del Papa, Maurizio Benucci, Marco Amedeo Cimmino, Valeria Riccieri, Fabrizio Conti, Gian Domenico Sebastiani, Annamaria Iuliano, Giacomo Emmi, Daniele Cammelli, Marco Sebastiani, Andreina Manfredi, Javier Bachiller-Corral, Walter Alberto Sifuentes Giraldo, Giuseppe Paolazzi, Lesley Ann Saketkoo, Roberto Giorgi, Fausto Salaffi, Jose Cifrian, Roberto Caporali, Francesco Locatelli, Enrico Marchioni, Alberto Pesci, Giulia Dei, Maria Rosa Pozzi, Lomater Claudia, Jorg Distler, Johannes Knitza, George Schett, Florenzo Iannone, Marco Fornaro, Franco Franceschini, Luca Quartuccio, Roberto Gerli, Elena Bartoloni, Silvia Bellando Randone, Giuseppe Zampogna, Montserrat I. Gonzalez Perez, Mayra Mejia, Esther Vicente, Konstantinos Triantafyllias, Raquel Lopez-Mejias, Marco Matucci-Cerinic, Albert Selva-O’Callaghan, Santos Castañeda, Carlomaurizio Montecucco, Miguel Angel Gonzalez-Gay

**Affiliations:** 1Department of Rheumatology, University and IRCCS Policlinico S. Matteo Foundation of Pavia and ERN ReCONNET, 27100 Pavia, Italy; gio.zanframundo@gmail.com (G.Z.); valentina.morandi02@universitadipavia.it (V.M.); francesco_locatelli@hotmail.it (F.L.); montecucco@smatteo.pv.it (C.M.); 2Department of Internal Medicine, Vall d’Hebron General Hospital, Universitat Autonoma de Barcelona, GEAS group, 08035 Barcelona, Spain; etraller@vhebron.net (E.T.-A.); aselva@vhebron.net (A.S.-O.); 3Department of Pneumology, University and IRCCS Policlinico S. Matteo Foundation of Pavia and ERN Lung, 27100 Pavia, Italy; f.meloni@smatteo.pv.it; 4Department of Rheumatology, University and ASST Spedali Civili—Brescia and ERN ReCONNET, 25123 Brescia, Italy; ilariacava@virgilio.it (I.C.); franco.franceschini1@gmail.com (F.F.); 5Interstitial Lung Disease and Rheumatology Unit, Instituto Nacional de Enfermedades Respiratorias, Ismael Cosio Villegas, 14080 Mexico City, Mexico; jorroser@gmail.com (J.R.-S.); ixchelglez19@gmail.com (M.I.G.P.); medithmejia1965@gmail.com (M.M.); 6Department of Rheumatology, Charité—Universitätsmedizin Berlin, 10117 Berlin, Germany; Eugen.Feist@helios-gesundheit.de (E.F.); Udo.Schneider@charite.de (U.S.); 7Department of Rheumatology, Hôpitaux Universitaires de Strasbourg and ERN ReCONNET, 67000 Strasbourg, France; alain.meyer1@chru-strasbourg.fr; 8Service de Physiologie des Explorations Fonctionnelles, NHC Strasbourg, Université de Strasbourg, 67000 Strasbourg, France; margherita.giannini85@gmail.com (M.G.); bernard.geny@chru-strasbourg.fr (B.G.); 9Department of Rheumatology, Centro Hospitalar e Universitário de Coimbra, 3000-075 Coimbra, Portugal; jdasilva@ci.uc.pt (J.A.P.d.S.); cjmc85@gmail.com (C.J.M.C.); 10Department of Rheumatology, Oslo University Hospital, 0372 Oslo, Norway; oyvind.molberg@medisin.uio.no (O.M.); helena.andersson@medisin.uio.no (H.A.); 11Department of Rheumatology, Cochin Hospital, 75014 Paris, France; veronicacodullo@yahoo.it; 12Department of Rheumatology, Azienda Ospedaliera Universitaria Pisana, Pisa and ERN ReCONNET, 56126 Pisa, Italy; marta.mosca@med.unipi.it (M.M.); simone.barsotti.pisa@gmail.com (S.B.); rneri@int.med.unipi.it (R.N.); 13Department of Rheumatology, Azienda Ospedaliero Universitaria S. Anna, 44124 Ferrara, Italy; carloalberto.scire@unife.it (C.S.); marcello.govoni@unife.it (M.G.); fefefurini@gmail.com (F.F.); 14Department of Rheumatology, Hospital General Universitario Gregorio Marañón, 28007 Madrid, Spain; fjlopezlongo@hotmail.com (F.J.L.-L.); juliamartinezbarrio@gmail.com (J.M.-B.); 15Department of Rheumatology, University of Heidelberg, 69117 Heidelberg, Germany; Hannes.Lorenz@med.uni-heidelberg.de; 16Department of Rheumatology, University of Padua and ERN ReCONNET, 35122 Padova, Italy; adoria@unipd.it (A.D.); anna.ghirardello@unipd.it (A.G.); 17Department of Rheumatology, Hospital Universitario San Cecilio, 18016 Granada, Spain; nortego@gmail.com; 18Department of Pneumology, University Hospital of Cattinara, 34149 Trieste, Italy; marco.confalonieri@asuits.sanita.fvg.it; 19Department of Rheumatology, University Hospital of Cattinara, 34149 Trieste, Italy; paolatomietto@gmail.com; 20Department of Rheumatology, S. Maria Hospital—IRCCS, 42123 Reggio Emilia, Italy; Nicolo.Pipitone@asmn.re.it; 21Department of Rheumatology, Severo Ochoa Hospital, 28911 Madrid, Spain; anabelen_r_c@hotmail.com (A.B.R.C.); mariblazquez@gmail.com (M.Á.B.C.); 22Department of Rheumatology and Clinical Immunology, Medical Center—University of Freiburg, Faculty of Medicine, University of Freiburg, 79110 Freiburg, Germany; reinhard.voll@uniklinik-freiburg.de (R.E.V.); sarah.wendel@uniklinik-freiburg.de (S.W.); 23Department of Rheumatology, Ospedale “ Scarlato” Scafati, 84018 Scafati, Italy; scarpasa@tin.it; 24Department of Rheumatology, HP Metz, Hopital Belle-Ile, 57000 Metz, France; francois.maurier@hp-metz.fr; 25Department of Rheumatology, ASST Papa Giovanni XXIII, 24127 Bergamo, Italy; mlimonta@asst-pg23.it; 26Department of Rheumatology, Ospedale di Treviglio, 24047 Treviglio, Italy; paolo_colombelli@asst-bgovest.it; 27Department of Rheumatology, Ospedale Guglielmo da Saliceto, 29121 Piacenza, Italy; e.arrigoni@ausl.pc.it (E.A.); bravielena@yahoo.it (E.B.); 28Department of Immunology, Azienda Ospedaliera Universitaria Pisana, Pisa and ERN ReCONNET, 56126 Pisa, Italy; paola.migliorini@med.unipi.it; 29Department of Rheumatology, University Clinic and AOU of Cagliari, 09100 Cagliari, Italy; mathieu@medicina.unica.it (A.M.); matteopiga@unica.it (M.P.); 30Department of Rheumatology, Johann Wolfgang Goethe-Universität, 60590 Frankfurt, Germany; ulrich.drott@kgu.de (U.D.); christiane.delbrueck@kgu.de (C.D.); 31Department of Rheumatology, ACURA Centre for Rheumatic Diseases, 76530 Baden-Baden, Germany; jutta.bauhammer@gmx.de; 32Department of Rheumatology, Skane University Hospital, 22242 Lund, Sweden; giovanni.cagnotto@med.lu.se; 33Department of Pneumology, AOU Catania, 95100 Catania, Italy; vancheri@unict.it (C.V.); dottorsambataro@gmail.com (G.S.); 34Department of Rheumatology, University Hospitals, 3000 Leuven, Belgium; ellen.delanghe@uzleuven.be; 35Department of Rheumatology at CAAD, DiMet, University of Eastern Piedmont (UPO) and AOU “Maggiore della Carità”, 28100 Novara, Italy; pierpaolo.sainaghi@med.uniupo.it; 36Department of Public Health, Unit of Biostatistics and Clinical Epidemiology, University of Pavia, 27100 Pavia, Italy; cristina.monti@unipv.it (C.M.); francesca.gigli@unipv.it (F.G.B.); 37Department of Rheumatology, Niguarda Hospital, 20162 Milan, Italy; mariaeva.romano@hotmail.it; 38Department of Pneumology, Ruhrlandklinik, University of Duisburg-Essen and ERN Lung, 45239 Essen, Germany; Francesco.Bonella@ruhrlandklinik.uk-essen.de; 39Department of Rheumatology, Ruhrlandklinik, University of Duisburg-Essen, 45239 Essen, Germany; ch.specker@gmail.com; 40Department of Rheumatology, Johannes Gutenberg-University, 55122 Mainz, Germany; schwarting@uni-mainz.de; 41Department of Rheumatology, Sierrallana Hospital, 39300 Torrelavega, Spain; villa.nacho@gmail.com; 42Department of Rheumatology, Humanitas Research Hospital, Rozzano, 20089 Milan, Italy; carlo.selmi@unimi.it (C.S.); dott.ceribelli@gmail.com (A.C.); 43Department of Rheumatology, Hospital Universitario La Paz, 28046 Madrid, Spain; lauranuno2@gmail.com; 44Department of Rheumatology, Hospital Clínico Universitario de Santiago de Compostela, 15702 Santiago de Compostela, Spain; antonio.mera.varela@sergas.es (A.M.-V.); nair.pg.89@gmail.com (N.P.G.); 45Department of Rheumatology, Città della Salute e della Scienza, 10126 Turin, Italy; fusaro.reumatorino@gmail.com (E.F.); simone.parisi@hotmail.it (S.P.); 46Department of Rheumatology, Hospital G. Pini—CTO, 20122 Milan, Italy; Luigi.Sinigaglia@asst-pini-cto.it (L.S.); Nicoletta.DelPapa@asst-pini-cto.it (N.D.P.); 47Department of Rheumatology, Azienda Ospedaliera San Giovanni di Dio, 50143 Firenze, Italy; maubenucci@tiscali.it; 48Department of Rheumatology, University of Genova 16126 Genova, Italy; cimmino@unige.it; 49Department of Rheumatology, University La Sapienza and Policlinico Umberto I, 00161 Rome, Italy; Valeria.Riccieri@uniroma1.it (V.R.); fabrizio.conti@uniroma1.it (F.C.); 50Department of Rheumatology, Ospedale San Camillo, 00152 Rome, Italy; giandoreum@libero.it (G.D.S.); annamariaiuliano@hotmail.it (A.I.); 51Department of Internal Medicine, AOU Careggi, 50134 Firenze, Italy; giacomaci@yahoo.it; 52Department of Immunology, AOU Careggi, 50134 Firenze, Italy; daniele.cammelli@unifi.it; 53Department of Rheumatology, Azienda Ospedaliera Universitaria di Modena, 41125 Modena, Italy; marco.sebastiani@unimore.it (M.S.); andreina.manfredi@gmail.com (A.M.); 54Department of Rheumatology, Hospital Universitario Ramon y Cajal, 28034 Madrid, Spain; javierbachiller@gmail.com (J.B.-C.); albertosifuentesg@gmail.com (W.A.S.G.); 55Department of Rheumatology, Ospedale Santa Chiara, 38122 Trento, Italy; Giuseppe.Paolazzi@apss.tn.it; 56University Medical Center- Comprehensive Pulmonary Hypertension Center & Interstitial Lung Disease Clinic Programs, Louisiana State University and Tulane University Schools of Medicine, Pulmonary Division New Orleans, New Orleans, LA 1542, USA; 57Department of Rheumatology, ASL Cuneo 2, 12051 Alba, Italy; rgiorgi@aslcn2.it; 58Department of Rheumatology, Polytechnic University of Marche, C. Urbani Hospital, 60035 Jesi, Italy; fausto.salaffi@gmail.com; 59Department of Pneumology, Hospital Universitario Marques de Valdecilla, IDIVAL, University of Cantabria Santander, 39008 Santander, Spain; jmcifrian@humv.es; 60Department of Clinical Sciences and Community Health, University of Milan and Gaetano Pini Hospital, 20122 Milan, Italy; roberto.caporali@unimi.it; 61Department of Neurology, IRCCS Mondino Foundation, 27100 Pavia, Italy; enrico.marchioni@mondino.it; 62Department of Pneumology, Univerity of Milano Bicocca, San Gerardo Hospital, 20900 Monza, Italy; alberto.pesci@unimib.it (A.P.); giulia.dei@hotmail.it (G.D.); m.pozzi@asst-monza.it (M.R.P.); 63Department of Rheumatology, Mauriziano Hospital, 10126 Turin, Italy; clomater@mauriziano.it; 64Department of Internal Medicine, Friedrich-Alexander-Universität Erlangen-Nürnberg, 91054 Erlangen, Germany; Joerg.Distler@uk-erlangen.de (J.D.); Johannes.Knitza@uk-erlangen.de (J.K.); georg.schett@uk-erlangen.de (G.S.); 65Rheumatology Unit—DETO, University of Bari, 70121 Bari, Italy; florenzo.iannone@uniba.it (F.I.); marco3987@hotmail.it (M.F.); 66Clinic of Rheumatology, Department of Medicine, Santa Maria della Misericordia Hospital and University of Udine, 33100 Udine, Italy; 67Rheumatology Unit, Department of Medicine, University of Perugia, 06129 Perugia, Italy; roberto.gerli@unipg.it (R.G.); elena.bartolonibocci@unipg.it (E.B.); 68Department of Rheumatology, AOU Careggi, 50134 Firenze, Italy; s.bellandorandone@gmail.com (S.B.R.); marco.matuccicerinic@unifi.it (M.M.-C.); 69Department of Rheumatology, AO Brunico, 39031 Bruneck, Italy; giuseppe.zampogna@sabes.it; 70Department of Rheumatology, Hospital Universitario de la Princesa, IIS-Princesa, 28006 Madrid, Spain; efvicenter@gmail.com (E.V.);; 71Department of Rheumatology, ACURA Center for Rheumatic Diseases, 55543 Bad Kreuznach, Germany; ktriantafyllias@gmail.com; 72Department of Rheumatology, Hospital Universitario Marques de Valdecilla, IDIVAL, University of Cantabria Santander, 39008 Santander, Spain; rlopezmejias78@gmail.com (R.L.-M.); miguelaggay@hotmail.com (M.A.G.-G.); 73Catedra UAM-Roche, EPID Future, Universitad Autonoma de Madrid, 28006 Madrid, Spain

**Keywords:** antisynthetase syndrome, antisynthetase antibodies, arthritis, myositis, interstitial lung disease

## Abstract

Antisynthetase syndrome (ASSD) is a rare clinical condition that is characterized by the occurrence of a classic clinical triad, encompassing myositis, arthritis, and interstitial lung disease (ILD), along with specific autoantibodies that are addressed to different aminoacyl tRNA synthetases (ARS). Until now, it has been unknown whether the presence of a different ARS might affect the clinical presentation, evolution, and outcome of ASSD. In this study, we retrospectively recorded the time of onset, characteristics, clustering of triad findings, and survival of 828 ASSD patients (593 anti-Jo1, 95 anti-PL7, 84 anti-PL12, 38 anti-EJ, and 18 anti-OJ), referring to AENEAS (American and European NEtwork of Antisynthetase Syndrome) collaborative group’s cohort. Comparisons were performed first between all ARS cases and then, in the case of significance, while using anti-Jo1 positive patients as the reference group. The characteristics of triad findings were similar and the onset mainly began with a single triad finding in all groups despite some differences in overall prevalence. The “ex-novo” occurrence of triad findings was only reduced in the anti-PL12-positive cohort, however, it occurred in a clinically relevant percentage of patients (30%). Moreover, survival was not influenced by the underlying anti-aminoacyl tRNA synthetase antibodies’ positivity, which confirmed that antisynthetase syndrome is a heterogeneous condition and that antibody specificity only partially influences the clinical presentation and evolution of this condition.

## 1. Background

Antisynthetase syndrome (ASSD) is a rare connective tissue disease that affects the skin, joints, muscles, and lungs [1]. Several specialists (e.g., neurologists, pulmonologists, and rheumatologists) may deal with this disease, and this wide range of potential referrals complicates ASSD treatment approaches and leads to heterogeneous management. The most important advances in ASSD have been mainly obtained in anti-Jo1 syndrome [2]. In this condition, the timing of the appearance of different manifestations is heterogeneous; isolated arthritis, generally similar to Rheumatoid Arthritis (RA), is the most frequent presentation [3,4,5,6], and anti-Ro antibodies co-occur in 50% of cases [7]. We have limited information regarding the clinical presentation pattern and evolution associated with other anti-aminoacyl tRNA synthetase antibodies (ARS). The aim of this study is to fully describe the effects of ARS in the clinical spectrum time course of ASSD and establish whether this condition is a unique syndrome or a group of different diseases.

## 2. Methods

### 2.1. Patients

According to the local Institutional Ethics Boards approval, the data were collected until the end of December 2018, from the AENEAS (American and European NEtwork of Antisynthetase Syndrome) collaborative group cohort of ASSD. The inclusion criteria were a clinical diagnosis of ASSD, being confirmed by ARS positivity, along with at least one triad finding. All of the patients should have been evaluated at least once in the local referral center between April 2014 (date of AENEAS collaborative group institution) and December 2018, with a follow-up of at least six months in order to be included in the study.

ASSD was defined complete (all triad findings) or incomplete (one/two triad findings) and the onset identified with the first pulmonary, muscle, or joint symptom/sign. Features’s onsets were defined as concomitant when they occurred less than three months apart. If a triad finding appeared more than three months after the previous one, it was defined as an “ex-novo” finding. Diagnostic delay was considered the time from the onset and the clinical diagnosis of ASSD.

### 2.2. Manifestation’s Definition

#### 2.2.1. Triad Findings

Interstitial lung disease (ILD): Restrictive pulmonary function tests (PFTs) pattern (forced vital capacity (FVC) ≤ 80%, forced expiratory volume in the first second/forced vital capacity (FEV1/FVC) ≥ 70%) and/or >20% diffusing capacity of the lungs for carbon monoxide (DLCO) reduction, and/or ground glass/reticular pattern on chest high-resolution computed tomography (HRCT). For HRCT scans, discussion with the local referent radiologist was mandatory, in order to reduce the risk of false-positive/negative patients. PFTs were performed at baseline, as an assessment of lung involvement in early arthritis/connective tissue disease, lung HRCT in the case of respiratory symptoms (cough and/or dyspnea), altered PFTs, DLCO impairment, or ASSD diagnosis. ILD presentation was defined as acute/subacute when dyspnea began acutely and progressed rapidly (4–6 weeks from symptom onset), chronic when dyspnea began insidiously and progressed slowly, and asymptomatic when lung involvement was not clinically evident.

Muscle involvement: Muscle enzymes’ elevation (creatinine phosphokinase and/or aldolase increase >50%, as compared with upper normal values) and typical electromyography and/or muscle biopsy and/or muscle magnetic resonance alterations. Myositis onset was defined as classic (muscle strength deficit) or hypomyopathic (no muscle strength deficit).

Arthritis: Clinical evidence of joint swelling/tenderness.

#### 2.2.2. Accompanying Findings

Fever: Body temperature of ≥38 °C for more than 10 days, not otherwise explained.

Mechanic’s hands (MHs): Thickened, hyperkeratotic, and fissured aspect of the radial sides of the fingers, without other explanations.

Raynaud’s Phenomenon (RP): Transient fingers’ ischemia after exposure to the cold, confirmed by a clinician.

### 2.3. Laboratory Tests

ARS were considered to be positive after two tests’ confirmation (at least one obtained in the tertiary autoimmune laboratory of the center that included the patient). The same “double positive” rule was applied for the antinuclear antibodies test (ANA test), and measures of rheumatoid factor (RF) and anti-cyclic citrullinated peptide antibodies (ACPA). The Euroline Autoimmune Inflammatory Myopathies 16 Ag kit (Euroimmun, Luebeck, Germany) was used in all centers for all cases of non-anti-Jo1 ARS, and for the majority of anti-Jo1 ARS cases (some of these were only assessed with commercially available ENA (extractable nuclear antigen) screen tests).

### 2.4. Statistical Analysis

The characteristics of the patients at disease onset and last follow-up were reported while using median and interquartile range (IQR) for the quantitative variables, and absolute/relative frequency values for the qualitative ones. The overall comparison among ARS groups was performed by a non-parametric Kruskal–Wallis test for quantitative variables, and the Chi-square or Fisher exact test for categorical variables, followed by post-hoc tests with Bonferroni corrected at α_c_ = 0.0125 (we applied the Mann–Whitney test for quantitative variables and Chi-square or Fisher exact test for categorical variables), while considering anti-Jo1 positive patients as the reference group. A Cox proportional hazard regression model was performed to evaluate the association between ARS type and the time of the “ex-novo” occurrence of triad findings, adjusted for sex, ASSD age at onset, and follow-up length. The Kaplan–Meier method and log-rank test were used to estimate survival and evaluate whether there are differences among ARS groups’ survival curves. Analyses were performed while using the STATA software package (2018, release 15.1; StataCorp, College Station, TX, USA).

## 3. Results

We retrospectively included 828 patients from 10 countries and 63 hospitals: 593 anti-Jo1 (72%), 95 anti-PL7 (11.5%), 84 anti-PL12 (10%), 38 anti-EJ (4.5%), and 18 anti-OJ (2%) ARS. Patients characteristics and comparisons by ARS group have been reported in the following paragraphs and in Table 1 (disease onset), Table 2 (last follow-up), Figure 1 (starting and final triad findings’ cluster), and Figure 2 (patients’ survival). Cox-regression for progression hazard ratio estimates, overtime “ex-novo” triad findings appearance, and treatment strategies have been included as Appendix A.

### 3.1. ARS Groups’ Characteristics

#### 3.1.1. Anti-Jo1 ARS

At onset, arthritis was the most common triad finding (362/593, 61%), being mainly polyarticular and symmetrical (253/354, 71.5%). Isolated arthritis was the main presentation form (*n* = 129/593, 22%). An incomplete ASSD was observed in 486/593 patients (82%); 302/486 incomplete ASSD patients (62%) developed “ex-novo” triad findings, at a median of 14 months (IQR 6–46) after disease onset. The most frequent “ex-novo” triad finding was ILD (187/294 patients without ILD at disease onset, 63%). Myositis (*n* = 487/593, 82%) and ILD (486/593, 82%) were the most common triad findings that were detected at the last follow-up. At that time, half of the patients had a complete ASSD (*n* = 298/593, 50%). The median follow-up of the 593 patients was 72 months (IQR 30–136). Sixty-five patients (11%) died (median 96 months, IQR 54–174 months after disease onset), and 86 (14%) were lost to follow-up (median 48 months, IQR 12–104 months after disease onset). Accompanying findings were reported in 416/587 (71%) patients (fever 211 patients, 36%; RP 216, 37%; MHs 217, 37%). In fact, 178/587 patients (30%) had more than one accompanying finding.

#### 3.1.2. Anti-PL7 ARS

At onset, ILD was the most common triad finding (52/95, 54%) and isolated ILD was the most common presentation form (32/95, 34%). An incomplete ASSD was observed in 86/95 patients (91%); 50/86 incomplete ASSD patients (58%) had “ex-novo” triad findings, at a median of 12 months (IQR 6–36), after disease onset. Myositis was the most common “ex-novo” (*n* = 30/49 patients without myositis at disease onset, 61%) and final (76/95, 80%) triad finding. At their last follow-up, 28/95 patients (29%) had a complete ASSD. Overall, the median follow-up length was 61 months (IQR 26–107). Twelve patients (13%) died (median 93 months, IQR 59–108 months after disease onset) and 6/95 (6%) were lost to follow-up (median 42 months, IQR 21–79 months after disease onset). Accompanying findings were reported in 70/92 (76%) patients (fever in 31 patients; 34%, RP in 46, 50%; MHs in 36, 42%). In fact, 37/92 patients (39%) had more than one accompanying finding.

#### 3.1.3. Anti-PL12 ARS

ILD was the most common triad finding (57/84, 69%) and isolated ILD was the main presentation form (36/84, 43%). An incomplete ASSD was observed in 79/84 patients (94%); 24/79 incomplete ASSD patients (30%) developed “ex-novo” triad findings at a median of 22 months (IQR 6–40), after disease onset. Myositis was the most frequent “ex-novo” triad finding (13/54 patients without myositis at disease onset, 24%). At their last follow-up, ILD (70/84, 83%) was again the most common triad finding and isolated ILD (*n* = 30, 36%) the main presentation form. The median follow-up of 84 patients was 38 months (IQR 20–75). Nine (11%) patients died (median 20 months, IQR 5–35 months after disease onset) and four (5%) were lost to follow-up (median 74 months, IQR 22–146, after disease onset). Accompanying findings were reported in 62/80 (78%) patients (fever in 33 patients, 53%; RP in 35, 56%; MHs in 28, 45%). In fact, 27/84 patients (32%) had more than one accompanying finding.

#### 3.1.4. Anti-EJ ARS

ILD was the most common triad finding (28/38, 74%) and isolated ILD the main presentation form (15/38, 39%). An incomplete ASSD was observed in 37 patients (97%); 23/37 incomplete ASSD patients (62%) had “ex-novo” triad findings, at a median of 12 months (IQR 6–24), after disease onset, mainly myositis (17/18 patients without myositis at disease onset, 94%). Most of the patients had complete ASSD at the end of follow-up (*n* = 15, 39%). The median follow-up of 38 patients was 36 months (IQR 17–106). Four patients died (11%, after median 24 months, IQR 18–74, follow-up) and five (13%) were lost to follow-up (13%, after median of 17 months, IQR 6–23 months, of follow-up). Accompanying findings were reported in 27 patients (71%, fever 14, 37%; RP e MHs 15, 39% respectively, *p* = 0.966). In fact, 40 patients (37%) had more than one accompanying finding.

#### 3.1.5. Anti-OJ ARS

Triad findings’ prevalence was not substantially different at disease onset (myositis, nine patients, 50%; ILD, eight patients, 44%; arthritis, seven patients, 41%); patients mainly presented with isolated myositis (*n* = 6, 33%) and an incomplete ASSD was observed in 17 cases (94%). Seven incomplete ASSD patients (41%) had “ex-novo” triad findings, at a median of 12 months (IQR 3–13), after disease onset. Myositis was the most common “ex-novo” (*n* = 5/9 patients without myositis at disease onset, 56%), and final (*n* = 14, 82%) triad finding. At their last follow-up, the patients showed isolated myositis (*n* = 5, 28%), myositis and ILD, or a complete ASSD (*n* = 4, 22% each). The median follow-up of 18 patients was 58 months (IQR 9–121). Only one patient was lost to follow-up, 58 months after disease onset. No patients died. Nine patients (50%) had accompanying findings (fever, one patient, 6%; RP, four patients, 22%; and, MHs, seven patients, 39%), even in association. Three patients (17%) had more than one accompanying finding.

### 3.2. Groups Comparisons: Main Results

In Table 1 and Table 2, we compared the main demographic, laboratory, single triad, and accompanying findings characteristics of our cohort. The diagnostic delay was greater in both anti-PL7 and anti-PL12 ARS (*p* < 0.001), as compared with anti-Jo-1 ARS, and the follow-up was shorter in anti-PL12 (*p* < 0.001) and anti-EJ ARS (*p* = 0.021). ANA test results (*p* > 0.05) and anti-Ro antibodies’ positivity were similar (*p* > 0.05). Anti-Jo1-positive patients had higher rates of arthritis when compared with other ARS (*p* < 0.01), with the exception of anti-OJ ARS, at onset (*p* = 0.06). When compared with anti-Jo-1 ARS, myositis was less common in anti-PL12 (*p* < 0.01), whereas ILD at disease onset was more common in anti-PL12 and anti-EJ (*p* < 0.01). Triad findings’ characteristics were similar, although, at the last follow-up, ILD presentation was mainly acute in anti-EJ (*p* = 0.003) as compared with anti-Jo-1 ARS. In Figure 1, we compare the cluster of triad findings according to underlying ARS, by first performing an overall comparison and then using the anti Jo-1 ARS as the reference group in the case of statistical significance (post-hoc analysis). At disease onset, isolated arthritis was the most common presentation form in anti-Jo1, an isolated ILD in anti-PL7, anti-PL12, and anti-EJ, and isolated myositis in anti-OJ ARS. At the last follow-up, the complete triad was the most common pattern in anti-Jo-1, anti-PL7, and anti-EJ; isolated ILD was the most common in anti-PL12, and isolated myositis in anti-OJ ARS. As shown in Table 2, as compared with anti-Jo1 ARS, “ex-novo” arthritis was less common in anti-PL12 and anti-OJ ARS (*p* < 0.01). “Ex-novo” myositis was only less common in anti-PL12 ARS (*p* < 0.001). The overtime progression of incomplete ASSD was only significantly reduced in anti-PL12 ARS (Appendix A). In fact, they had 58% less “risk” of progression, when compared with anti-Jo1 ARS (Appendix A: Hazard Ratio = 0.42, 95% confidence interval (CI) 0.28–0.65), with the same sex, age of onset, and follow-up length, being considered as follow-up intervals (0–12 months, 13–24 months, 25–60 months, and >60 months). Survival was not different between groups, as the Kaplan–Maier curve, as reported in Figure 2, shows.

## 4. Discussion

Despite recent advances [3,5,6,7,8,9,10,11], many questions on ASSD are still unsolved. A major problem is the lack of established classification criteria [12,13], with the subsequent inclusion of ASSD patients with a wide spectrum of conditions, such as Interstitial Pneumonia with Autoimmune Features [14,15], polymyositis/dermatomyositis [16,17], idiopathic pulmonary fibrosis [2], and RA [2,3,7,18]. By considering this unmet need, a conjoint international team is working to establish the ACR-EULAR Classification Criteria of ASSD. However, the first step before the beginning of this project is to clearly define the clinical spectrum time course that is associated with ARSs, such as anti-PL7, PL12, OJ, and EJ (defined as non-anti-Jo1 ARS), and not only those with anti-Jo1 antibodies, in order to confirm that ASSD shares several characteristics.

To date, only a few studies have focused on the comparison between anti-Jo1 and non-anti-Jo1 ARS. The most important [8] identified two different clusters: the first, including the anti-Jo1 ARS, was associated with multi-organ involvement, the second, including anti-PL7 and anti-PL12 ARS, was mainly lung-limited. These results were confirmed in other cohorts [9,19]. Other authors suggested a different timing of myositis and ILD appearance, and an ARS-related heterogeneity of the syndrome [17]. However, these studies only partially focused on arthritis, a frequently forgotten manifestation of ASSD, at risk of misdiagnosis with RA [3,7]. We showed that, even if arthritis was substantially more frequent in anti-Jo1 positive patients (about 75%), the characteristics were similar independently to the underlying ARS specificity. Furthermore, non-anti-Jo1 ARS patients had arthritis at a clinically relevant rate, in about 40%–50% of cases. However, we cannot exclude a possible underestimation of non-anti-Jo1 ARS in isolated arthritis [3]. In fact, although anti-ENA screen tests, usually containing anti-Jo1 antibodies, are routinely performed on these patients, non-anti-Jo1 ARS are rarely tested. Muscle involvement was less common in anti-PL12 ARS, but the characteristics were similar in all groups. The final prevalence of ILD was also similar and the only difference that we observed was that anti-EJ positive patients more frequently had an acute onset. These similarities and the similar prevalence of ANA test and anti-Ro antibodies’ positivity could be considered a confirmation of the similar nature of ASSDs. Another clue relates to a similar triad findings cluster and clinical spectrum time course, which we first confirmed in non-anti-Jo1 ARS. A single triad finding at onset (isolated-arthritis in anti-Jo1 ARS, isolated-myositis in anti-PL7 and anti-OJ ARS, isolated-ILD in anti-PL12 and anti-EJ ARS) was the most common onset type, and the “ex-novo” occurrence of previously lacking triad findings in incomplete ASSD was similar in all of the groups. Only anti-PL12 positive patients had less progression but, in this case, the clinical meaning largely outweighed the statistical result, since the 30% rate of progression in incomplete ASSD appears to be one relevant event. Lastly, we also compared survival, which was substantially similar. Our result is in contrast with some previous reports [8,9,20], but in keeping with others [19,21]. We do not know the reasons for these differences, but we feel that early diagnosis and referral of ASSD may improve prognosis.

The main limitation of this study is its retrospective design [22,23]. However, we strongly believe that this type of study is a necessary starting point in all clinical research, in particular if addressing rare diseases. Furthermore, the large number of patients included in the study might partially compensate for its retrospective nature. In some patients, anti-Jo1 ARS positivity was only ascertained through routine ENA screening tests. Many of these patients had died or were lost to follow-up, thus they did not have the possibility of being re-tested with the reference kit used in the local tertiary laboratories. We should question whether patients with anti-Jo1 antibodies should perform additional analyses other than the routine ENA screening tests in a real-life setting. The same problem led us to not include the characterization of anti-Ro (52 or 60 kDA) antibodies. Another potential limitation is that we did not use immunoprecipitation (IP) for ARS positivity confirmation, and that we did not evaluate anti-Zo, -YRS, and -KS ARS. However, when considering that IP cannot be routinely applied in daily clinical practice [24], this could be considered to be a cohort of patients from a real-life setting. Furthermore, our choice to only include patients with twice-confirmed ARS positivity might have excluded some ASSD patients from analysis. However, we preferred losing some true ASSD cases rather than increase the risk of false-positive patients’ inclusion. Additionally, the mandatory determination of ARS in a tertiary laboratory centre was used to get the cleanest population study possible. Lastly, the cytoplasmic positivity of ANA [11] was not available in a relevant percentage of cases, and thus not included in this analysis. Regarding the ILD definition that we applied, it is important to underline that the accuracy/robustness of the radiological diagnoses has not been centrally verified, and that the patterns of ILD have not been characterized in this study.

## 5. Conclusions

In conclusion, our data strongly suggest that the clinical presentation and course of anti-Jo1, PL7, PL12, EJ, and anti-OJ positive ASSD is broadly similar, regardless of the specific antibody responsible, sharing various characteristics and triad feature type, ANA and anti-Ro positivity, prevalence of accompanying findings, and having a similar clinical spectrum time [7]. This is not a minor issue, because several ARS-positive patients are not diagnosed with ASSD [2]. Obviously, different diagnoses correspond to different approaches, with changes in the treatment strategies and potential effects on the outcome. We think that the present study is a further confirmation of the clinical need for specific classification criteria for ASSD, which will provide a basis for setting up specific clinical therapeutic trials for this disease.

## Figures and Tables

**Figure 1 jcm-08-02013-f001:**
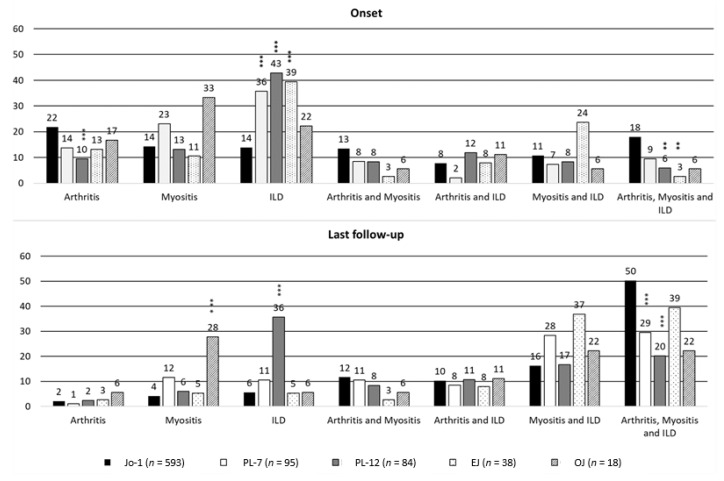
Triad findings cluster (in percentage) at disease onset and last follow-up. Significant differences between groups were evaluated considering anti-Jo1 positive patients as the reference group. Chi-square or Fisher exact test were used as appropriate (significance threshold: *p* < 0.0125, reported as *** *p* < 0.001, ** *p* < 0.01). Legend: ILD, interstitial lung disease.

**Figure 2 jcm-08-02013-f002:**
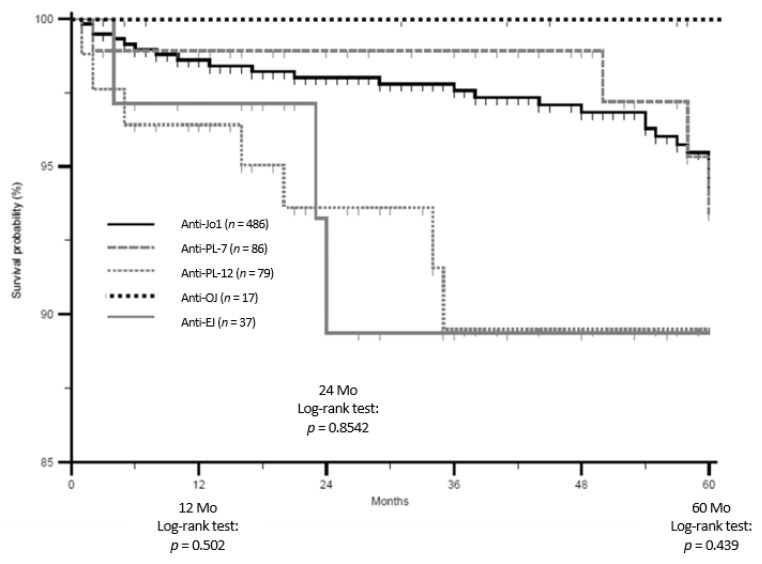
The Kaplan–Meier survival curve in our cohort of antisynthetase syndrome patients (established range: 0–60 months) with Log-rank test at 12, 24 and 60 months. Mo: months.

**Table 1 jcm-08-02013-t001:** Characteristics of included patients at disease onset.

Onset Characteristics	Anti-Jo-1 ARS (*n* = 593)	Anti-PL-7 ARS (*n* = 95)	Anti-PL-12 ARS (*n* = 84)	Anti-EJ ARS (*n* = 38)	Anti-OJ ARS (*n* = 18)	Test; *p*-Value; d*f*
**^§^ Females (%)**	433 (73.1)	71 (74.7)	62 (73.8)	29 (76.3)	13 (72.2)	χ2 = 0.29; 0.99; 4
**Median age in years at disease onset (IQR)**	51.0 (41.0–61.0)	53.0 (44.0–63.0)	50.5 (42.5–61.5)	54.5 (46.0–62.0)	57.0 (47.0–67.0)	^χ2 = 5.09; 0.28; 4
**Median diagostic delay in months (IQR)**	5.0 (2.0–15)	12.0 (5.0–41.0)	10.0 (4.0–26.0)	6.0 (2.0–12.0)	8.0 (2.0–58.0)	**^χ2 = 41.60; 0.0001; 4**
comparison vs Anti-Jo-1 ARS	reference	***** ***p* < 0.0001**	***** ***p* = 0.0002**	*p* = 0.95	*p* = 0.10	
**ANA positive (%)**	350 (60.3)	58 (64.4)	49 (59.4)	21 (60.00)	7 (38.9)	χ2 = 4.14; 0.39; 4
**ANA negative (%)**	230 (39.7)	32 (35.6)	34 (41.0)	14 (40.0)	11 (61.1)	
**Anti Ro positive (%)**	301 (51.3)	50 (54.3)	44 (59.2)	19 (50.0)	4 (22.2)	χ2 = 8.32; 0.08; 4
**Anti Ro negative (%)**	286 (48.7)	42 (45.7)	31 (40.8)	19(50.0)	14 (77.8)	
**Arthritis (%)**	362 (61.1)	32 (33.7)	30 (35.7)	10 (26.3)	7 (38.9)	**χ2 = 52.02; <0.0001; 4**
comparison vs Anti-Jo1-1 ARS	reference	***** ***p* < 0.0001**	***** ***p* < 0.0001**	***** ***p* < 0.0001**	*p* = 0.06	
**Symmetrical polyarthritis (%)**	253 (71.5)	21 (65.6)	20 (71.4)	8 (80.0)	2 (28.6)	Fisher exact test *p* = 0.20
**Oligoarticular/asymmetrical arthritis (%)**	101 (28.5)	11 (34.4)	8 (28.6)	2 (20.0)	5 (71.4)	
**IgM-RF positive (%)**	91 (26.2)	9 (32.1)	5 (19.2)	4 (40.0)	3 (50.0)	Fisher exact test *p* = 0.39
**IgM-RF negative (%)**	256 (73.8)	19 (67.9)	21 (80.8)	6 (60.0)	3 (50.0)	
**ACPA positive (%)**	34 (11.2)	5 (20.8)	1 (3.8)	2 (25.0)	0 (0)	Fisher exact test *p* = 0.20
**ACPA negative (%)**	270 (88.8)	19 (79.2)	25 (96.2)	6 (75.0)	6 (100)	
**Myositis (%)**	336 (56.7)	46 (48.4)	30 (35.7)	15 (39.5)	9 (50.0)	**χ2 = 16.87; 0.002; 4**
comparison vs Anti-Jo-1 ARS	reference	*P* = 0.13	***** ***p* < 0.0001**	*p* = 0.04	*p* = 0.57	
**Hypomyopathic onset (%)**	51 (15.2)	12 (26.1)	9 (30.0)	3 (20.0)	3 (33.3)	Fisher exact test *p* = 0.06
**Classic onset (%)**	284 (84.8)	34 (73.9)	21 (70.0)	12 (80.0)	6 (67.7)	
**Interstitial Lung Disease (%)**	299 (50.4)	52 (54.7)	57 (69.0)	28 (73.7)	8 (44.4)	**χ2 = 17.29; 0.002; 4**
comparison vs Anti-Jo-1 ARS	reference	*p* = 0.43	***** ***p* = 0.001**	***** ***p* = 0.005**	*p*= 0.62	
**Acute onset (%)**	132 (44.6)	25 (48.1)	28 (48.3)	20 (74.1)	3 (37.5)	Fisher exact test *p* = 0.14
**Chronic onset (%)**	114 (38.5)	18 (34.6)	25 (43.1)	6 (22.2)	4 (50.0)	
**Asymptomatic onset (%)**	50 (16.9)	9 (17.3)	5 (8.6)	1 (3.7)	1 (3.7)	

Statistically significant differences in bold. * Post-hoc tests significance threshold: *p* < 0.0125. Legend: ARS, antisynthetase antibodies; IQR, interquartile range; ANA, antinuclear antibodies; RF, Rheumatoid factor; ACPA, anticyclic citrullinated peptide antibodies; ^χ2, Kruskal–Wallis test; d*f*, degree of freedom. ^§^ 1 subject in the sample is transgender.

**Table 2 jcm-08-02013-t002:** Characteristics of included patients at last follow-up.

Last Follow-Up Characteristics	Anti-Jo-1 ARS (*n* = 593)	Anti-PL-7 ARS (*n*= 95)	Anti-PL-12 ARS (*n* = 84)	Anti-EJ ARS (*n* = 38)	Anti-OJ ARS (*n*= 18)	Test; *p*-Value; d*f*
**Median disease duration in months (IQR)**	72 (30–136)	61 (26–107)	37.5 (20–73.5)	35.5 (17–102)	57.5 (9–118)	**^χ2 = 19.29; 0.0007; 4**
comparison vs Anti-Jo-1 ARS	reference	0.320	***<** **0.001**	0.021	0.334	
**Arthritis (%)**	440 (74.2)	47 (49.5)	35 (41.7)	20 (52.6)	8 (44.4)	**χ2 = 58.54; <0.001; 4**
comparison vs Anti-Jo-1 ARS	reference	***<** **0.001**	***<** **0.001**	***** **0.004**	***** **0.005**	
**Symmetrical polyarthritis (%)**	294 (69.0)	31 (66.0)	23 (69.7)	14 (77.8)	3 (37.5)	χ2 = 4.50; 0.342; 4
**Oligoarticular/asymmetrical arthritis (%)**	132 (31.0)	16 (34.0)	10 (30.3)	4 (22.2)	5 (62.5)	
**IgM-RF positive (%)**	107 (25.4)	12 (28.6)	5 (16.7)	7 (36.8)	3 (42.9)	Fisher exact test *p* = 0.392
**IgM-RF negative (%)**	314 (74.6)	30 (71.4)	25 (83.3)	12 (63.2)	4 (57.1)	
**ACPA positive (%)**	37 (10.3)	8 (22.2)	2 (6.4)	2 (15.4)	0 (0.0)	Fisher exact test *p* = 0.185
**ACPA negative (%)**	323 (89.7)	28 (77.8)	29 (93.6)	11 (84.6)	7 (100.0)	
**Patients with X-rays joint erosions (%)**	58 (15.3)	4 (11.4)	1 (4.3)	2 (20.0)	2 (28.6)	Fisher exact test *p* = 0.360
**Patients without X-rays joint erosions (%)**	320 (84.7)	31 (85.6)	22 (95.6)	8 (80.0)	5 (71.4)	
**Myositis (%)**	487 (82.1)	76 (80.0)	43 (51.2)	32 (84.2)	14 (77.8)	**χ2 = 42.93; <0.001; 4**
comparison vs Anti-Jo-1 ARS	reference	0.62	***<** **0.001**	0.636	0.74	
**Hypomyopathic onset (%)**	97 (20.0)	23 (30.7)	12 (27.9)	6 (18.7)	5 (35.7)	χ2 = 7.01; 0.135; 4
**Classic onset (%)**	388 (80.0)	52 (69.3)	31 (72.1)	26 (81.3)	9 (64.3)	
**Interstitial Lung Disease (%)**	486 (82.0)	73 (76.8)	70 (83.3)	34 (89.5)	11 (61.1)	χ2 = 8.16; 0.086; 4
**Acute onset (%)**	179 (37.3)	28 (38.9)	34 (48.6)	21 (63.6)	4 (36.4)	**Fisher exact test *p* < 0.001**
**Chronic onset (%)**	201 (41.9)	32 (44.4)	28 (40.0)	11 (33.3)	6 (54.6)	
**Asymptomatic onset (%)**	100 (20.8)	12 (16.7)	8 (11.4)	1 (3.0)	1 (9.1)	
comparison vs Anti-Jo-1 ARS	reference	0.713	0.09	***** ***p* = 0.003**	*p* = 0.623	
**Incomplete ASSD with** **“** **ex-novo** **”** **triad findings (%)**	302 (62.1)	50 (58.1)	24 (30.4)	23 (62.2)	7 (41.2)	**χ2 = 30.27; <0.001; 4**
**Incomplete ASSD without** **“** **ex-novo** **”** **triad findings (%)**	184 (37.9)	36 (41.9)	55 (69.6)	14 (37.8)	10 (58.8)	
comparison vs Anti-Jo-1 ARS	reference	0.482	***<** **0.001**	0.998	0.081	
**“ex** **-novo** **”** **arthritis (%)**	78 (33.8)	15 (23.8)	5 (9.3)	10 (35.7)	1 (9.1)	**χ2 = 16.48; 0.002; 4**
**no arthritis (%)**	153 (66.2)	48 (76.2)	49 (90.7)	18 (64.3)	10 (90.9)	
comparison vs Anti-Jo-1 ARS	reference	0.13	***<** **0.001**	0.837	***0.005**	
**“ex-** **novo** **”** **myositis (%)**	151 (58.8)	30 (61.2)	13 (24.1)	17 (73.9)	5 (55.6)	**χ2 = 26.43; <0.001; 4**
**no myositis (%)**	106 (41.3)	19 (38.8)	41 (75.9)	6 (26.1)	4 (44.4)	
comparison vs Anti-Jo-1 ARS	reference	0.75	***<** **0.001**	0.155	*p* = 1.00	
**“ex-** **novo** **”** **Interstitial lung disease (%)**	187 (63.6)	21 (48.8)	12 (46.2)	6 (60.0)	3 (30.0)	**χ2 = 9.63; 0.047; 4**
**no Interstitial lung disease (%)**	107 (36.4)	22 (51.2)	14 (53.9)	4 (40.0)	7 (70.0)	
comparison vs Anti-Jo-1 ARS	reference	0.063	0.079	0.816	0.031	
**Accompanying features (%)**	416 (70.9)	70 (76.1)	62 (77.5)	27 (71.1)	9 (50.0)	χ2 = 6.58; 0.160; 4
**No accompanying findings (%)**	171 (29.1)	22 (23.9)	18 (22.5)	11 (29.0)	9 (50.0)	

Statistically significant differences in bold. * Post-hoc tests significance threshold: *p* < 0.0125. Legend: ARS, antisynthetase antibodies; IQR, interquartile range; RF, Rheumatoid factor; ACPA, anticyclic citrullinated peptide antibodies; ^χ2, Kruskal–Wallis test; d*f*, degree of freedom.

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
