# Peer review of "Influence of Antisynthetase Antibodies Specificities on Antisynthetase Syndrome Clinical Spectrum Time Course"

_jcm, 2019, doi:10.3390/jcm8112013_

Round 1
Reviewer 1 Report
This is a multicenter retrospective study to describe the characteristics of different anti-aminoacyl tRNA synthetase (ARS) antibody subgroups and to identify whether antisynthetase syndrome (ASSD) is a unique disease or a group of different diseases. The study deals with an interesting topic and new data about this rare ASSD may provide an important contribution in real-world clinical settings. However, I have serious concerns about the methodology.
It is well-known that commercial line blot kits for detection of myositis-specific autoantibodies (MSA) including anti-ARS antibody may be useful as a screening test in the clinical setting but provide false-positive results. Reliable confirmation of anti-ARS antibody-positivity is the key of this study. Therefore, in my opinion, the kits are not suitable for this study to identify the clinical significance of anti-ARS antibodies. Immunoprecipitation assay (IP), rather than immunoblotting method, is a gold standard method to detect MSAs. It is mandatory to confirm the positivity of anti-ARS antibody by IP. The authors should include and re-analyze only patients with anti-ARS antibody confirmed by IP. Multiple pairwise group comparisons were performed in this study. In statics, multiple comparison is associated with some problems including false positives/type I errors. Therefore, multiple-testing corrections are needed. Page 6, line 268: the definition of "diagnostic delay" was not described in the Methods section.Author Response
Dear Reviewer
First of all, I want to say thanks for your suggestions. Every point you raised has been assessed and we tried to explain in the better way possible the aspects that we did not report clearly. We really hope that you could consider satisfactory our answers.
Regarding your suggestions:
Comment: It is well-known that commercial line blot kits for detection of myositis-specific autoantibodies (MSA) including anti-ARS antibody may be useful as a screening test in the clinical setting but provide false-positive results. Reliable confirmation of anti-ARS antibody-positivity is the key of this study. Therefore, in my opinion, the kits are not suitable for this study to identify the clinical significance of anti-ARS antibodies. Immunoprecipitation assay (IP), rather than immunoblotting method, is a gold standard method to detect MSAs. It is mandatory to confirm the positivity of anti-ARS antibody by IP. The authors should include and re-analyze only patients with anti-ARS antibody confirmed by IP. Multiple pairwise group comparisons were performed in this study. In statics, multiple comparison is associated with some problems including false positives/type I errors. Therefore, multiple-testing corrections are needed
Answer: we agree that commercially available kits for antisynthetase antibodies (ARS) have well-defined problems and that immunoprecipitation is the gold standard for the determination of ARS. However, this gold standard is not generally applied in the daily routine practice, because not worldwide diffuse, not easy to do and expensive. In general practice, the identification of ARS is based on commercially available kits. Sincerely, we cannot report the results of immunoprecipitation, because we had not enough funding for this purpose and, we obtained the IRB approval only for the retrospective analysis of data. As in our previous paper (Sebastiani M, J Rheumatol 2019; Gonzalez-Gay MA, Clin Exp Rheumatol 2018; Scirè CA, Respir Med 2017; Bartoloni-Bocci E, Autoimmunity Rev 2017; Cavagna L, CRAI 2017; Cavagna L, Medicine 2015), we described in this paper patients from the real world, where clinicians have not access to immunoprecipitation, but only to commercially available kits, with the advantages and defects of these kits. We tried to reduce the problems, by avoiding the inclusion of double ARS positive patients, and by including only patients with a clinical diagnosis of antisynthetase syndrome satisfying well-established inclusion criteria. Substantially, this is a cohort of real-live antisynthetase syndrome patients. The results we reported are the ones observed in the routine practice and thus useful for clinicians. Multiple-testing problem was considered and the post-hoc analyses used a Bonferroni-corrected significance threshold as explained in the modified text of the statistical analysis section and also it is declared in table 1, 2 and figure 1 legends. According to both reviewers suggestions, we modified methods, results, and discussion. Our text changes have been reported in yellow.
We hope that this explanation could be enough for the reviewer and that the clinical strength of the paper could overweight the lack of immunoprecipitation. We modified some parts of the text by underlying that patients had a clinical diagnosis of antisynthetase syndrome (thus reducing the risk of false-positive results) and by adding the lack of immunoprecipitation as a limit of the study, that we did not evaluate some ARS specificities and we modified statistical part.
Methods: Inclusion criteria were the clinical diagnosis of ASSD by an expert clinician, confirmed by ARS positivity along with at least one triad finding. Methods: Statistical analysis: Overall comparison among ARS groups were performed by non-parametric Kruskal-Wallis test for quantitative variables and Chi-square or Fisher exact test for categorical variables, followed by post-hoc tests with Bonferroni corrected αc=0,0125 (we applied Mann-Whitney test for quantitative variables and Chi-square or Fisher exact test for categorical variables), considering anti-Jo1 positive patients as the reference group. Discussion: Another potential limitation is that we did not use immunoprecipitation (IP) for ARS positivity confirmation and that we did not evaluate anti-Zo, -YRS, and –KS ARS. However, by considering that IP cannot be routinely applied in the daily clinical practice, this could be considered a cohort of patients from a real live setting.
Comment: Page 6, line 268: the definition of "diagnostic delay" was not described in the Methods section.
Answer: we added the definition of diagnostic delay in the Methods section, In particular in Methods: Diagnostic delay was considered the time from the onset and the clinical diagnosis of ASSD.
Reviewer 2 Report
This is an interesting manuscript that will definitely make an important contribution to the field. The comprehensive characterization of anti-synthetase syndrome is welcome, and the large number of patients considered, across a wide geographical base, is impressive. The data on temporal accumulation of clinical features across different antibody-specific categories was also very useful clinically, and I think this will make a very valuable reference point. The authors are also open about some key limitations, particularly the retrospective nature of the study. I am therefore very positive overall, but there are a number of concerns about the manuscript the paper. If these were addressed, the reader would find the paper easier to understand.
MAJOR COMMENTS
I suspect the statistical analysis is not ideal. When applying a Kruskal-Wallis test (for example) my understanding is that the authors should use a post-hoc test rather than multiple head to head tests comparing anti-Jo1 with each of the other antibody types. Assuming they did head to head tests, it is not clear what they used. Perhaps they did do the appropriate post-hoc test (the legends describe a post-hoc test, but not which one). This raises the concern that the statistical analysis has not been performed by someone experienced in statistics, and this should be checked. The statistics should be revised accordingly. I suspect the overall findings and message of the paper will be very similar, but they should still be achieved with the appropriate statistical methods. While it is good that the authors openly accept limitations of a retrospective design, they should also answer a few concerns in the methods. For example, how far back did the analysis go? Was there a target sample size before the study started - if so what was it and how was it calculated?; if not, how did they decide when to stop? Was there a MINIMUM duration of follow up required for entry into the study? Did they get institutional ethics approval for ALL sites? What was the funding source for the study? Was the composition of the groups and the definition of the clinical parameters pre-determined (i.e. before the data were collected), or were these decided upon after the data were collected? I assume they were pre-determined, but do the authors have any evidence for this? This type of information should be provided. Following on from this, what criteria were used to define interstitial lung disease? Were all scans reviewed by a study radiologist or were they reports from issuing hospitals? If these were based on old reports from host hospitals, it will be important to have a section in the discussion to say (a) that the accuracy/robustness of the radiological diagnoses has not been verified and (b) that the patterns of ILD have not been characterised in this study. Both of these are limitations, and the paper should say so. There is a high chance that this approach will have yielded false positives (i.e. cases where there is not true ILD) and also that it will have missed some "true" cases of ILD. Were all cases of ILD made on HRCT? Any patients who did not have an HRCT cannot, by definition, have been diagnosed with ILD, and this will have led to missed cases. On the other hand, if the authors HAVE had all scans checked by a study thoracic radiologist(s) with an interest in ILD, it would be very interesting if they reported the patterns of disease, in this or another publication. The authors make the good point that their (very appropriate) requirement for 2 positive anti-ARS tests (at least one in the reference lab for the study) will have led to the exclusion of many potential patients. If they have a feel for approximately how many patients may have been missed because of this approach, they should give an indication (this is the same argument as "if an HRCT was not done, you cannot diagnose an ILD"). This introduces an important bias, and I would recommend that they emphasise this limitation even more strongly in the revision. I also think they should indicate where the reference lab was. Furthermore, they only report 5 anti-ARS antibodies (with the implication that no patients had >1 anti-ARS antibody), but they make no mention of whether other, rarer anti-ARS antibodies were ever tested (e.g. anti-KS, anti-Zo etc). This should be commented on. I personally felt that Figure 2 was poorly presented, and distracted from the very interesting tables and Figures 1 and 3. Most of the key points in Fig 2 can be inferred from Fig 1. I would remove Fig 2 or move it to the Supplementary Section.MINOR COMMENTS
In the conclusions, I feel it is too strong to say that they have "confirmed" that ASSD is a "unique entity". I think it would be more balanced to say that their data strongly suggest that the clinical presentation and course of ASSD is broadly similar, regardless of the specific responsible antibody (or words to that effect). The English is generally good, but there are some places where poor English makes the paper hard to understand, and a few places where key points are diluted. Examples include the last sentence in the "Patients" section; the use of ex novo (it would be much more helpful to the reader to use English instead of Latin); the use of the word "instrumental" (I still am not certain I know what they mean by this); and, perhaps most importantly, the title of the manuscript does not make good sense and should be clearer (clinically and grammatically). There are several more examples, and I strongly recommend that they show a revision to an author or clinical colleague whose first language is English. In my copy the line colours in Figure 3 made it hard to differentiate some of the groups.Author Response
Dear Reviewer
thanks a lot for your suggestions. We think that the paper is steadily improved in quality and the limits are better defined. Please note that in the text, every change has been highlighted in yellow.
Major suggestions
Comment: I suspect the statistical analysis is not ideal. When applying a Kruskal-Wallis test (for example) my understanding is that the authors should use a post-hoc test rather than multiple head to head tests comparing anti-Jo1 with each of the other antibody types. Assuming they did head to head tests, it is not clear what they used. Perhaps they did do the appropriate post-hoc test (the legends describe a post-hoc test, but not which one). This raises the concern that the statistical analysis has not been performed by someone experienced in statistics, and this should be checked. The statistics should be revised accordingly. I suspect the overall findings and message of the paper will be very similar, but they should still be achieved with the appropriate statistical methods
Answer: Thanks to the reviewer for pointing out some doubts. Statisticians worked together with the clinicians to define and to apply the correct statistical analyses plan from study beginning. It was our intention first to evaluate differences among ARS spectrum, and then, in case of difference confirmation, to compare anti-Jo1 positive patients with other ARS specificities. To this latter purpose, we performed 4 post-hoc tests (Jo-1 vs PL7, vs PL12, vs OJ, vs EJ) for different variables. We agree that our explanation was not so clear as needed. Thus, in order to better define the statistical methods that have been applied, we have modified the statistical analysis section and the results section, by explaining the post-hoc methods we applied. For example, when we compared quantitative variables among ARS groups, the statistical analysis was carried out with the purpose of analyzing the variability of median values of each group with respect to the median value of anti-Jo1. Therefore, after Kruskal-Wallis test, we didn’t do a k(k-1)/2 head to head post hoc testing (where k indicates the number of groups), but we performed (k-1) Mann Whitney post hoc test considering anti-Jo1 as the reference group. The significance threshold for every post hoc Mann-Whitney’s test was corrected to alpha/(k-1). We modified the text as follow. Statistical analysis: Overall comparison among ARS groups were performed by non-parametric Kruskal-Wallis test for quantitative variables and Chi-square or Fisher exact test for categorical variables, followed by post-hoc tests with Bonferroni corrected αc=0,0125 (we applied Mann-Whitney test for quantitative variables and Chi-square or Fisher exact test for categorical variables), considering anti-Jo1 positive patients as the reference group. Groups comparisons: main results In Table 1 and 2 we compared the main demographic, laboratory, single triad and accompanying findings characteristics of our cohort. The diagnostic delay was greater in both anti-PL7 and anti-PL12 ARS (p<0.001) with respect to anti-Jo-1 ARS, as well as follow-up was shorter in anti-PL12 (p<0.001) and in anti-EJ ARS (p=0.021), ANA test (p>0.05) and anti-Ro antibodies positivity were similar (p>0.05). Anti-Jo1 positive patients had more frequently arthritis with respect to other ARS (p <0.01) with the exception of anti-OJ ARS at the onset (p=0.06). With respect to anti-Jo-1 ARS, myositis was less common in anti-PL12 (p<0.01) whereas ILD at disease onset was more common in anti-PL12 and anti-EJ (p<0.01). Triad findings’ characteristics were similar, although at last follow-up ILD presentation was mainly acute in anti-EJ ARS (p=0.003) with respect to anti-Jo-1 ARS. In figure 1 we compared the cluster of triad findings according to underlying ARS, by performing first an overall comparison and then by using the anti-Jo-1 ARS as the reference group in case of statistical significance (post-hoc analysis).
Comment: how far back did the analysis go? Was there a target sample size before the study started - if so what was it and how was it calculated? If not, how did they decide when to stop? Was there a MINIMUM duration of follow up required for entry into the study? Did they get institutional ethics approval for ALL sites? What was the funding source for the study? Was the composition of the groups and the definition of the clinical parameters pre-determined (i.e. before the data were collected), or were these decided upon after the data were collected? I assume they were pre-determined, but do the authors have any evidence for this? This type of information should be provided.
Answer: we included patients with antisynthetase syndrome that were evaluated at least once between 2014 and 2018 in participating centers (added in the text). Data collection beginning is concomitant with the establishment of the AENEAS collaborative group. We had not a sample size at the beginning, we asked to collect all patients available and we obtained a very incredible result. We limited the analysis to patients with at least 6 months of follow-up (added in the text). We involved only centers with IRB approval, that allowed a retrospective data analysis. We reported the funding sources (FOREUM grant) after the conclusions. Funding sources were used for a data manager in Pavia. Regarding the group composition, the majority of centers participated from the beginning to AENEAS collaborative group, however, some centers were added subsequently, the last one in 2017. Clinical parameter collection was predetermined. The evidence for both groups and clinical parameters is clearly evidenced in our previous papers (Sebastiani M, J Rheumatol 2019; Gonzalez-Gay MA, Clin Exp Rheumatol 2018; Scirè CA, Respir Med 2017; Bartoloni-Bocci E, Autoimmunity Rev 2017; Cavagna L, CRAI 2017; Cavagna L, Medicine 2015).
Comment: following on from this, what criteria were used to define interstitial lung disease? Were all scans reviewed by a study radiologist or were they reports from issuing hospitals? If these were based on old reports from host hospitals, it will be important to have a section in the discussion to say (a) that the accuracy/robustness of the radiological diagnoses has not been verified and (b) that the patterns of ILD have not been characterized in this study. Both of these are limitations, and the paper should say so. There is a high chance that this approach will have yielded false positives (i.e. cases where there is not true ILD) and also that it will have missed some "true" cases of ILD. Were all cases of ILD made on HRCT? Any patients who did not have an HRCT cannot, by definition, have been diagnosed with ILD, and this will have led to missed cases. On the other hand, if the authors HAVE had all scans checked by a study thoracic radiologist(s) with an interest in ILD, it would be very interesting if they reported the patterns of disease, in this or another publication. The authors make the good point that their (very appropriate) requirement for 2 positive anti-ARS tests (at least one in the reference lab for the study) will have led to the exclusion of many potential patients. If they have a feel for approximately how many patients may have been missed because of this approach, they should give an indication (this is the same argument as "if an HRCT was not done, you cannot diagnose an ILD"). This introduces an important bias, and I would recommend that they emphasize this limitation even more strongly in the revision. I also think they should indicate where the reference lab was. Furthermore, they only report 5 anti-ARS antibodies (with the implication that no patients had >1 anti-ARS antibody), but they make no mention of whether other, rarer anti-ARS antibodies were ever tested (e.g. anti-KS, anti-Zo etc). This should be commented on. I personally felt that Figure 2 was poorly presented, and distracted from the very interesting tables and Figures 1 and 3. Most of the key points in Fig 2 can be inferred from Fig 1. I would remove Fig 2 or move it to the Supplementary Section.
Answer: Interstitial lung disease was defined as in our previous studies: restrictive pulmonary function tests (PFTs) pattern (FVC≤80%, FEV1/FVC≥70%) and/or>20% DLCO reduction, and/or ground glass/reticular pattern on chest high-resolution computed tomography (HRCT). Substantially by reviewing data, all patients had HRCT evidence of ILD, however, by considering the starting variables definition (in 2014) we did not change the inclusion criteria. We added that the for the inclusion it was not enough a report describing ILD occurrence, but it was necessary that clinicians re-evaluated with the local reference radiologist the occurrence (or not) of ILD. In the text, we added: Methods: For HRCT scans it was mandatory the discussion with the local referent radiologist, in order to reduce the risk of false-positive/negative patients. Unfortunately, we did not review centrally the HRCT scans, because the transfer of a so large number of HRCT scan was not so easy and because we had not the IRB approval for images exchange. Sincerely, we have some indirect supporting data, as for example the concurrent occurrence of DLCO reduction and PFTs signs of restriction in the majority of enclosed patients (we did not collect these values but only if they were impaired or not). Furthermore (and sincerely), now we have an ongoing Italian study on electronic evaluation of HRCT scans in ASSD. We have 200 HRCT of patients that we included in the present study and all patients had consistent findings of ILD according to our referent radiologist. We will discuss the HRCT pattern in another paper. Substantially we completely agree with this (and other) comment(s) and we added some points in the text and we moved figure 2 in the supplemental materials. As suggested, in the Discussion: Regarding ILD definition we applied, it is important to underline that the accuracy/robustness of the radiological diagnoses has not been centrally verified and that the patterns of ILD have not been characterized in this study.
Comment: The authors make the good point that their (very appropriate) requirement for 2 positive anti-ARS tests (at least one in the reference lab for the study) will have led to the exclusion of many potential patients. If they have a feel for approximately how many patients may have been missed because of this approach, they should give an indication (this is the same argument as "if an HRCT was not done, you cannot diagnose an ILD"). This introduces an important bias, and I would recommend that they emphasize this limitation even more strongly in the revision. I also think they should indicate where the reference lab was.
Answer: Unfortunately, we have not this information, because we excluded these patients from data collection according to our inclusion criteria, established in 2014. Furthermore, I can only report the Pavia experience. Substantially, I’m following 72 antisynthetase patients and from 2014 I did not confirm the positivity of antisynthetase antibodies not tested in my laboratory in 15 cases. These patients, furthermore, were not clinically diagnosed with antisynthetase syndrome (7 systemic sclerosis, 5 Sjogren syndrome, 3 mixed connective tissue disease). We are really sorry that we cannot satisfy reviewer request, however, we modified the text by putting the attention on the pitfall indicated in the comment. Furthermore, we did not explain well that we did not perform a central ARS positivity confirmation, and that final ARS positivity was always confirmed in the tertiary laboratory of the center that included the patients. According to other reviewer suggestions, we modified the text, in particular, in the part Laboratory test we modified as follow: (at least one obtained in the tertiary autoimmune laboratory of the center that included the patient. Always in the part Laboratory tests: The Euroline Autoimmune Inflammatory Myopathies 16 Ag kit (Euroimmun, Luebeck, Germany) was used in all centers for all non-anti-Jo1 ARS. In the Discussion, we added: Our choice to include only patients with twice-confirmed ARS positivity may have excluded from the analysis of some ASSD. However, we preferred to lose some true ASSD than increase the risk of false-positive patients’ inclusion. Also, the mandatory determination of ARS in a tertiary laboratory center was tried to get the cleanest population study possible. Of note, according to first reviewer suggestion, we indicated that we did not perform immunoprecipitation and that this is another bias of the study.
Comment: Furthermore, they only report 5 anti-ARS antibodies (with the implication that no patients had >1 anti-ARS antibody), but they make no mention of whether other, rarer anti-ARS antibodies were ever tested (e.g. anti-KS, anti-Zo etc). This should be commented on. I personally felt that Figure 2 was poorly presented, and distracted from the very interesting tables and Figures 1 and 3. Most of the key points in Fig 2 can be inferred from Fig 1. I would remove Fig 2 or move it to the Supplementary Section
Answer: unfortunately, we have not the possibility to evaluate the occurrence of additional anti-ARS. The commercial kit used kit applied in different centres did not include these antibodies. In the purpose of our study, we indicated the type of ARS antibodies we included in the study. We added in the text that we focused on specific ARS (those enclosed in the kit used in different laboratory centres) and the lack of some ARS in our study, in particular, in the first part of the Discussion: However, the first mandatory step before the beginning of this project is to clearly define the clinical spectrum time course associated with other ARS (anti-PL7, PL12, OJ and EJ, defined as non-anti-Jo1 ARS), and not only with anti-Jo1 antibodies, to confirm that ASSD is effectively a unique entity.
Always in the Discussion, we added: Another potential limitation is that we did not use immunoprecipitation (IP) for ARS positivity confirmation and that we did not evaluate anti-Zo, -YRS, and –KS ARS. However, by considering that IP cannot be routinely applied in the daily clinical practice, this could be considered a cohort of patients from a real live setting.
Minor suggestions:
Comments: In the conclusions, I feel it is too strong to say that they have "confirmed" that ASSD is a "unique entity". I think it would be more balanced to say that their data strongly suggest that the clinical presentation and course of ASSD is broadly similar, regardless of the specific responsible antibody (or words to that effect). The English is generally good, but there are some places where poor English makes the paper hard to understand, and a few places where key points are diluted. Examples include the last sentence in the "Patients" section; the use of ex novo (it would be much more helpful to the reader to use English instead of Latin); the use of the word "instrumental" (I still am not certain I know what they mean by this); and, perhaps most importantly, the title of the manuscript does not make good sense and should be clearer (clinically and grammatically). There are several more examples, and I strongly recommend that they show a revision to an author or clinical colleague whose first language is English. In my copy the line colours in Figure 3 made it hard to differentiate some of the groups.
Answers: We changed the conclusions in line with the reviewers’ suggestion. The phrase reported by the reviewer reported is perfect. We reassessed English, but we are asking if we can maintain the term ex-novo. We used this term from the first AENEAS publication. If it is possible, we will be happy, also for our (ancient roman) ancestry. We explained in patients section the meaning of ex-novo: If a triad finding appeared more than 3 months after the previous one, it was defined as “ex-novo” finding. Also, instrumental problems have been fixed. Finally, we modified the title of the paper: Influence of antisynthetase antibodies specificities on antisynthetase syndrome clinical spectrum time course. We modified figure 3 (now 2), in order to better differentiate ARS groups.
Round 2
Reviewer 1 Report
I appreciate the authors’ response. The manuscript has been revised well. I think this manuscript will be acceptable after some corrections have been done.
Page 4, line 169: Is “the time from the onset and the clinical diagnosis of ASSD” correct? I presume that “the time from the onset until the clinical diagnosis of ASSD” is correct. Page 6, line 278-: the authors often used “with respect to anti-Jo-1 ARS” in this paragraph. Does this mean “compared with anti-Jo-1 ARS” ? Figure 1: Although “*p>0.05” was described in figure 1 legend, it does not appear in Figure 1. The manuscript would benefit from language editing by either a native English speaker or a professional editor.
Author Response
We are very grateful to the Reviewer.
We performed asked changes.
We requested the English revision to the system
Best Regards
Lorenzo Cavagna and Miguel Angel Gonzalez-Gay